# Behaviour of Human Oral Epithelial Cells Grown on Invisalign^®^ SmartTrack^®^ Material

**DOI:** 10.3390/ma13235311

**Published:** 2020-11-24

**Authors:** Michael Nemec, Hans Magnus Bartholomaeus, Michael H. Bertl, Christian Behm, Hassan Ali Shokoohi-Tabrizi, Erwin Jonke, Oleh Andrukhov, Xiaohui Rausch-Fan

**Affiliations:** 1Division of Orthodontics, University Clinic of Dentistry, Medical University of Vienna, 1090 Vienna, Austria; michael.nemec@meduniwien.ac.at (M.N.); michael.bertl@meduniwien.ac.at (M.H.B.); christian.behm@meduniwien.ac.at (C.B.); erwin.jonke@meduniwien.ac.at (E.J.); 2Division of Conservative Dentistry and Periodontology, University Clinic of Dentistry, Medical University of Vienna, 1090 Vienna, Austria; hans.bartholomaeus@meduniwien.ac.at (H.M.B.); hassan.shokoohi-tabrizi@meduniwien.ac.at (H.A.S.-T.); xiaohui.rausch-fan@meduniwien.ac.at (X.R.-F.)

**Keywords:** orthodontics, aligner, proliferation, epithelial barrier, inflammation, oral epithelial cells, in vitro, Invisalign, SmartTrack

## Abstract

Invisalign aligners have been widely used to correct malocclusions, but their effect on oral cells is poorly known. Previous research evaluated the impact of aligners’ eluates on various cells, but the cell behavior in direct contact with aligners is not yet studied. In the present study, we seeded oral epithelial cells (cell line Ca9-22) directly on Invisalign SmartTrack material. This material is composed of polyurethane and co-polyester and exhibit better mechanical characteristics compared to the predecessor. Cell morphology and behavior were investigated by scanning electron microscopy and an optical cell moves analyzer. The effect of aligners on cell proliferation/viability was assessed by cell-counting kit (CCK)-8 and 3,4,5-dimethylthiazol-2-yl-2,5-diphenyl tetrazolium bromide (MTT) assay and live/dead staining. The expression of inflammatory markers and proteins involved in epithelial barrier function was measured by qPCR. Cells formed cluster-like structures on aligners. The proliferation/viability of cells growing on aligners was significantly lower (*p* < 0.05) compared to those growing on tissue culture plastic (TCP). Live/dead staining revealed a rare occurrence of dead cells on aligners. The gene expression level of all inflammatory markers in cells grown on aligners’ surfaces was significantly increased (*p* < 0.05) compared to cells grown on TCP after two days. Gene expression levels of the proteins involved in barrier function significantly increased (*p* < 0.05) on aligners’ surfaces after two and seven days of culture. Aligners’ material exhibits no cytotoxic effect on oral epithelial cells, but alters their behavior and the expression of proteins involved in the inflammatory response, and barrier function. The clinical relevance of these effects has still to be established.

## 1. Introduction

In the past decade, there has been a significant increase in esthetic concerns of orthodontic patients regarding orthodontic appliances. This led to an intensified use and development of aligner treatments [1,2]. Aligners are an orthodontic treatment modality, consisting of a series of removable clear plastic appliances, which should be worn 22 h per day [3]. Depending on the treatment regime, patients should switch to the next aligner every week or two weeks, resulting in a gradual correction of the patients’ malocclusion [4].

The world market leader in aligner treatment is Align Technology and its product Invisalign^®^, with over 6 million aligner treatments (https://www.invisalign-professional.de/). In 2013 the precursor material Exceed30 [5] was replaced by the latest SmartTrack^®^ material. SmartTrack aligners are made of polyurethane and co-polyester, whereas Exceed30 aligners are made from polyurethane methylene diphenyldiisocyanate 1,6-hexanediol [5]. Both aligners are made by the thermoforming process, but compared to Exceed30, SmartTrack demonstrates greater elastic recovery and smaller residual deformation [5]. Aligners are produced based on 3D printed casts and consist of a flat surface on the outside and a rough surface on the inside [3].

As aligners should only be removed for food intake and oral hygiene, the aligners remain most of the time in the oral cavity and could, therefore, exert possible effects on the oral health homeostasis [6]. Reported clinical side effects of aligner treatment are the development of white spot lesions [7], decreased periodontal status, and increased levels of periodontopathic bacteria due to generally impaired oral hygiene during orthodontic treatment [8,9]. Biocompatibility and application material safety are essential factors, which should be considered during aligner therapy. However, a recent meta-analysis on safety considerations of orthodontic aligners revealed that the current evidence is inconsistent, and further laboratory and clinical studies are needed [10]. A potential cytotoxic effect of aligners’ eluates was tested in several in vitro studies, reporting rather weak cytotoxic activity of aligner eluates toward gingival fibroblasts, keratinocytes, and breast cancer cells [11,12,13].

As oral epithelium fulfills a major role in protecting the underlying structures from bacterial and physical insults [14,15], particular attention should be devoted to avoid any detrimental effects of Invisalign material on this tissue. To the best of our knowledge, only one study investigates the impact of SmartTrack material on oral cells [11]. This study found slight toxic effects of SmartTrack’ eluates on human primary gingival fibroblasts. Since aligners are situated in direct contact with oral epithelium during their application [11], it would be important to explore the behavior of epithelial cells directly grown on aligners.

Hence, this in vitro study aimed to examine the cell functional and morphological parameters of human oral squamous carcinoma cells directly grown on aligners made of SmartTrack material. Oral squamous carcinoma cells share many properties of normal oral epithelial cells and were used as the model of the oral epithelium [16]. Cell morphology, proliferation/viability, cell death, and gene expression of several functional proteins were investigated. Since integrins and E-cadherin play a pivotal role in the inter-relation of epithelial cells with the extracellular matrix [17,18], we focused on these parameters to investigate cell barrier function. Inflammatory cytokines, such as interleukin (IL)-8, a strong factor associated with chemotaxis, and IL-6 and intercellular adhesion molecule 1 (ICAM-1) [19,20], both associated with inflammatory conditions, were also investigated in this study.

## 2. Materials and Methods

To obtain aligners with flat surfaces, the buccal areas of conventional orthodontic plaster models have been ground plane and one set of aligners (Invisalign, Align Technology, San Jose, CA, USA) containing 69 upper and lower aligners was ordered. Aligners made from SmartTrack material were manufactured by Align Technology (San Jose, CA, USA). In the region of these plane buccal surfaces of aligners, discs of 6 mm in diameter were cut out with a revolving punch plier (Knippex, Wuppertal, Germany), mainly in the region of upper incisors and upper molars (Figure 1). The required diameter of flat aligner surface was confirmed by using a conventional orthodontic caliper (Zürcher Model 125 mm, Karl Hammacher GmbH, Solingen, Germany). Aligner specimens have been UV sterilized twice for further testing. Aligners have two different surfaces: a rough inner and a smooth outer surface [21]. Cells were grown on both, inner (tooth-facing) and outer surfaces of aligners and cultured up to seven days. Surface hydrophobicity was characterized by the contact angle measured by the sessile drop technique as described previously [22,23].

### 2.1. Cell Culture

We used the commercially available human oral squamous carcinoma cell line Ca9-22 (Japanese Collection of Research Bioresources Cell Bank, JCRB0625, Ibaraki, Japan). Ca9-22 cells were cultured in modified Eagle’s minimum essential medium (MEM, Gibco^®^, Carlsbad, CA, USA), supplemented with 10% fetal bovine serum, penicillin (100 U/mL), and streptomycin (50 μg/mL) at 37 °C in a humidified atmosphere containing 5% CO_2_. Cells between the fourth and the seventh passage were used for the experiments.

The discs were fixed in 96-well cell culture plates using colorless high-vacuum silicone grease (Sigma Aldrich, St. Louis, MO, USA). Our preliminary experiment showed no effect of silicon grease on Ca9-22 cells (data not shown), which is similar to our previous finding on other cell type (osteoblast-like MG-63 cells, [24]). 5 × 10^3^ cells, re-suspended in 15 µL of MEM supplemented with 10% fetal bovine serum, penicillin (100 U/mL), and streptomycin (50 μg/mL) were seeded on the discs. Cells were cultured on both the inner and outer surface of aligners. After 4 h, 85 µL of MEM with supplements was added to each well, and cells were further cultured for either two or seven days. As control, 5 × 10^3^ cells were seeded in 15 µL medium on tissue culture plastic (TCP) which was followed by adding further 85 µL medium after 4 h.

### 2.2. Cell Morphological and Functional Parameters

#### 2.2.1. Scanning Electron Microscopy

The morphology and microstructure of Ca9-22 cells grown on inner and outer SmartTrack surfaces were analyzed using scanning electron microscopy (SEM, Quanta 200, FEI, Hillsboro, OR, USA); [25]). Cells were seeded on aligners’ discs and cultured at 37 °C as described above. Specimens in each group were scanned under SEM at two and seven days, similarly to the previously described method [25]. For SEM, the specimens were fixed with 4% formaldehyde for at least 24 h and washed three times with phosphate-buffered saline (PBS) to remove unattached cells. After the specimens were dehydrated by rinsing with gradually increased ethanol (Merck, Darmstadt, Germany), ethanol was exchanged by hexamethyldisilazane (Sigma-Aldrich, St. Louis, MO, USA) and specimens were coated with a 100 nm thin gold layer using a sputter coater (EM ACE200, Leica, Wetzlar, Germany). Specimens were observed under the SEM at an accelerating voltage of 15 kV and at 400- and 1500-fold magnification. The SEM images of cross-sectional and surface views were acquired. SEM analysis was performed in triplicates for each type of preparation.

#### 2.2.2. Live Cell Analyser

Imaging of cell growth was performed by JuLi™ Br live-cell movie analyzer (Nanoentek, Seoul, Korea), a bright-field optical system, for seven days. Time-lapse images were captured every 30 min at 37 °C in a humidified atmosphere containing 5% CO_2_ by 4x magnification at a cell concentration of 5 × 10^3^ in 96 well plates. Time-lapse analysis was repeated three times for each condition.

#### 2.2.3. Cell Proliferation/Viability

Cell proliferation/viability was tested by using 3,4,5-dimethylthiazol-2-yl-2,5-diphenyl tetrazolium bromide (MTT) dye (Sigma Aldrich, St. Louis, MO, USA) and cell counting kit 8 (CCK-8, Dojindo Laboratories, Kumamoto, Japan), as described in previous studies [24,26]. Ca9-22 cells were seeded on aligners or in wells of 96 well plates, as described above. After two and seven days incubation, 30 μL of MTT solution (5 mg/mL in PBS) were added per well, followed by incubation at 37 ° C for 4 h. After discarding conditioned media, 150 µL dimethyl sulfoxide (Merck, Darmstadt, Germany) were added per well, followed by 5 min incubation. Finally, 80 μL of each cultured solution were transferred to a 96-well plate, and the optical density [27] was measured at 570 nm with a photometer (Synergy HTX, Biotek, Winooski, VT, USA).

For CCK-8 experiments, 10 μL CCK-8 reagent were added per well followed by incubation at 37 °C for 4 h. 80 μL of each well were transferred to a 96-well plate, and the optical density was measured at 450 nm [28] using a photometer (Synergy HTX, Biotek, VT, USA). Experiments were performed at least in triplicates.

#### 2.2.4. Live/Dead Staining

Viability of Ca 9-22 cells was assessed by live/dead staining using the Live/Dead Cell Staining Kit (Enzo Life Sciences, Lausen, Switzerland) following the manufacturer’s guidelines [29]. Live/dead analysis was performed after seven days of culture. Cell staining was visualized using a fluorescent microscope (Nikon, Tokyo, Japan; excitation wavelength: 488 nm; emission wavelength living cells: 518 nm, green; emission wavelength dead cells: 615 nm, red).

#### 2.2.5. Quantitative Real-Time PCR (qPCR)

The expression levels of different functional proteins in Ca9-22 cells were quantified using qPCR after two and seven days of culture, similarly to the previously described method [30]. Lysate preparation, mRNA transcription into cDNA, and qPCR were performed using the TaqMan^®^ Gene Expression Cells-to-CT™ kit (Ambion/Applied Biosystems, Foster City, CA, USA) according to the manufacturer’s instructions. Reverse transcription was performed using Primus 96 advanced thermocycler (Applied Biosystems, Foster City, CA, USA). qPCR was performed on an ABI StepOnePlus device (Applied Biosystems, Foster City, CA, USA) in paired reactions using TaqMan^®^ gene expression assays with the following ID numbers (all from Applied Biosystems): Interleukin (IL)-6, Hs00985639_m1; Interleukin (IL)-8, Hs00174103_m1; integrin alpha 6 (ITG-α6), Hs01041011_m1; integrin beta 4 (ITG-β4), Hs00173995_m1; E-cadherin, Hs01023894_m1; glyceraldehyde 3-phosphate dehydrogenase (GAPDH), Hs99999905_m1. GAPDH was used as a house-keeping gene. The PCR reactions were performed in triplicate under the following conditions: 95 °C for 10 min; 50 cycles, each for 15 s at 95 °C and at 60 °C for 1 min. For each sample, the point at which the PCR product was first detected above a fixed threshold was determined as the cycle threshold (Ct). The 2^−ΔΔCt^ method was used to calculate the relative expression of target genes: ΔΔCt = (C_t_^target^ − C_t_^GAPDH^)_sample_ − (C_t_^target^ − C_t_^GAPDH^) _vehicle control_. Experiments were performed at least in triplicates.

### 2.3. Statistical Analysis

The normal distribution of all data was tested with the Kolmogorov–Smirnov test. Statistical differences were determined using a one-way analysis of variance (ANOVA) for paired data followed by a post-hoc LSD test for pairwise comparison. All analyses were performed using SPSS 23.0 (SPSS Inc, Chicago, Illinois, IL, USA). The level of statistical significance was set at 0.05. All data are expressed as mean ± S.E.M of four independent experiments.

## 3. Results

### 3.1. Contact Angle

The contact angle measured by the sessile drop technique was found to be (mean ± s.e.m., n = 5) 87.3 ± 2.3 for TCP, 91.9 ± 1.9 for the inner aligner’s surface, and 84.6 ± 1.6 for the outer aligner’s surface. The contact angle for the inner surface was significantly lower than that for the outer surface; no significant difference between TCP and both surfaces was detected.

### 3.2. Scanning Electron Microscopy

Figure 2 shows representative SEM images taken after 48 h of incubation at different magnifications. At lower magnification, Ca9-22 cells were uniformly distributed on the TCP and formed cluster-like structures on aligners. The cluster size and the number of cells included tended to be higher for the inner surface than the outer surface. SEM images at higher magnification showed that the cells within the clusters also form direct contact with the surface of aligners.

### 3.3. Live Cell Analyzer

Figure 3 shows representative pictures of Ca9-22 cells grown on different surfaces. Photos were taken after two and seven days of culture. Cells on TCP grew continuously and formed a confluent layer after seven days. In contrast, Ca9-22 cells grown on aligners’ surfaces formed some cluster-like structures. No visible alteration in the size and numbers of cell clusters was observed during the incubation period. The size of clusters was larger for Ca9-22 grown on the inner surface compared to those grown on the outer surface of aligners.

### 3.4. Proliferation/Viability of Ca9-22 Cells

The proliferation/viability of Ca9-22 cells grown on different surfaces for two and seven days was measured by MTT and CCK-8 and is shown in Figure 4. Both assays show that the proliferation/viability of Ca9-22 cells grown on aligners’ surfaces was significantly lower compared to those grown on TCP. Cells grown on the inner surface exhibited slightly lower proliferation/viability compared to the outer surface after both two and seven days of culture. However, a statistically significant difference between outer and inner surfaces was detected only by the MTT method (*p* < 0.05), but not by the CCK-8 method.

### 3.5. Live/Dead Staining

Figure 5 shows representative pictures of live/dead staining of Ca9-22 cells grown on different surfaces for seven days. Despite the formation of clusters, most cells grown on aligners’ surfaces retain viability during the incubation time. The occurrence of dead cells was relatively rare.

### 3.6. Gene Expressions of Inflammatory Markers

The gene expression levels of IL-6, IL-8, and ICAM-1 in Ca9-22 cells grown on the different surfaces for two and seven days are shown in Figure 6. After two days, the gene expression level of all inflammatory markers in cells grown on aligners’ surfaces was significantly increased compared to cells grown on TCP. The gene expression level of IL-6 was significantly higher in cells grown on the inner surface than the outer surface, whereas no significant differences in IL-8 and ICAM-1 expression were observed between inner and smooth surfaces. After seven days, the gene expression levels of IL-6 and ICAM-1 in cells grown on aligners’ surfaces were still significantly higher compared to the cells grown on TCP. In contrast, the gene expression level of IL-8 was lower on aligners’ surfaces than that on TCP. Furthermore, the gene expression of IL-8 in Ca9-22 cells grown on the inner aligners’ surface was significantly lower compared to the outer surface after seven days. No significant differences in the gene expression of IL-6 and ICAM-1 were observed between inner and outer surfaces.

### 3.7. Gene Expression Levels of Factors Involved in the Barrier Function

The effect of different aligners’ surfaces on the expression of ITG-α6, ITG-β4, and E-cadherin is shown in Figure 7. Gene expression levels of all three parameters significantly increased on both aligners’ surfaces after two and seven days (*p* < 0.05). All parameters showed no statistically significant differences between different aligners’ surfaces (*p* > 0.05).

## 4. Discussion

The main advantage of clear aligner therapy over conventional multibracket orthodontics is the lack of bonded components. It is well known that fixed orthodontic appliances can alter oral microbiome [31], whereas clear aligner therapy has minimal effects on the mouth environment [32]. However, the exact effect of clear aligners in the oral cavity has still to be investigated. To date, several papers are already published concerning the chemical and mechanical properties of different aligners’ materials [33,34,35,36]. In contrast to numerous studies on aligner treated patients, there are a relatively low number of studies investigating the cellular effects of SmartTrack material [11,12,13]. These studies mainly report the impact of aligners’ eluates on cell viability/proliferation. However, to the best of our knowledge, no study investigated the behavior and functional characteristics of cells grown directly on aligners. Therefore, in the present study, we investigated the behavior of oral epithelial cells grown on aligners’ surfaces. Particularly, we focused on the cell proliferation/viability, morphology, and the expression of various proteins involved in the epithelial barrier function and local inflammatory response.

An important observation of our study is that epithelial cells do not grow on SmartTrack aligner material. This was proved by assessing the proliferation/viability of epithelial cells using the MTT and CCK-8 methods. We have not seen any substantial differences in cell proliferation/viability between two and seven days of culture on aligners. In contrast, cells on TCP showed markedly higher proliferation/viability after seven days of culture compared to two days of culture. This observation suggests the poor growing state of epithelial cells on aligners. This conclusion is further supported by microscopic analysis. Both SEM and live cells analyzer showed higher cell density after seven days compared to two days for TCP. Live/dead staining showed a low proportion of dead cells on both aligners’ surfaces and TCP.

A previous study with aligners’ eluate reported a decrease in proliferation/viability of oral immortalized epithelial cells only by maximal 21.2 ± 6.35% [11]. This effect is substantially lower than that observed in our study. The reason for this difference could be that the surface of aligners is not suitable for cell growth. This assumption is also supported by SEM analysis, in which epithelial cells grown on the aligners formed clusters. Only the cells on the border of these clusters are directly bound to the surface. In contrast, cells within the clusters prefer to interact with other cells and not with the surface of the aligner.

All surfaces exhibited similar contact angles. Therefore the fact that cells growing on aligners exhibit low proliferation and different attachment and behavior compared to TCP could be explained by the different composition of these materials. SmartTrack material is composed of aromatic thermoplastic polyurethane and co-polyester. In contrast, TCP is usually made from polystyrene and is additionally treated with extracellular matrix components to facilitate cell attachment and growth. A previous study suggests that polyurethane-based surface is not optimal for cell culture [37]. Particularly, the growth of different cell types, including keratinocytes, is reported to be decreased for polyurethane compared to TCP [37]. Polyurethane exhibited an impaired attachment of osteosarcoma derived MG-63 cells [38]. However, previous studies did not find a cytotoxic effect of polyurethane on the different cell types [39]. Thus, the altered behavior of Ca9-22 cells on SmartTrack material can be explained by polyurethane material, but the role of other components should be further investigated.

Maintenance of oral epithelial barrier is vital to protect the host from the environment, pathogens, exogenous substances and mechanical stress. Intercellular junctions and hemi-desmosomes are essential components of the epithelial barrier and can be found in the junctional epithelium [14,40]. This tissue displays the connective part between tooth surface and gingiva, mediating the connection between hard and soft tissue in the oral cavity [41]. A significant role within the maturation and formation of these intercellular junctions plays E-cadherin, a transmembrane protein. Hemi-desmosomes are comprised of integrins, a family of transmembrane receptor proteins. ITG-α6 and ITG-β4 subunits play an essential role in connecting the oral epithelium to the basal membrane [17].

Hence, to assess the potential effect of aligners’ material on the epithelial barrier function, we have investigated the gene expression of E-cadherin, ITG-α6, and ITG-β4. Epithelial cells grown on aligners exhibited significantly higher gene expression levels of E-cadherin and integrin subunits than TCP. This effect was more pronounced after two days compared to seven days of culture. The increase in the expression of proteins involved in the intercellular interaction could be linked to the different cells’ behavior on aligners and TCP. Mainly, an increased expression of E-cadherin and integrin subunits in cells growing on aligners could be associated with an increased cluster formation. Simultaneously, our data do not imply any detrimental effect of aligners’ material on the epithelial barrier function. Noteworthy, oral squamous carcinoma cells generally exhibit lower E-cadherin expression compared to the healthy oral epithelium [42]. This fact should be considered by the data interpretation.

The oral epithelium is not only a mechanical barrier but also actively participates in the oral cavity’s inflammatory processes through the production of different proteins. IL-6 and IL-8, both inflammatory mediators, are known to be involved in the progression of periodontal disease [43,44]. We have investigated these inflammatory mediators because of their role in the development of gingivitis [45]. Gingivitis is a common side effect of orthodontic therapy [46] and can further develop into the more severe condition of periodontitis. Intercellular adhesion molecule (ICAM)-1 is expressed by vascular endothelium of gingival blood vessels and junctional epithelium [47]. Further, it is known that ICAM-1 is expressed by junctional epithelial cells in healthy gingiva and by pocket epithelium in diseased gingiva, but on the contrary, not detectable on most keratinocytes in the external gingival epithelium [48]. ICAM-1 expression in oral epithelial cells is increased under inflammatory conditions [49,50,51].

The expression of all inflammatory parameters was substantially increased in the epithelial cells grown on aligners compared to TCP. However, a different time course for various inflammatory parameters was observed. Whereas the expression of IL-8 and ICAM-1 was markedly higher after two days compared to seven days, an opposite dependency for IL-6 was found. IL-8 is a known chemokine and is the main factor regulating transepithelial neutrophil migration [52,53]. IL-8 alteration can affect neutrophil activity, and ICAM-1 is involved in the leukocyte adhesion in oral epithelial cells [50,54]. Thus, both IL-8 and ICAM are involved in the regulation of oral host-microbiome interaction. Therefore, increased expression of these parameters by aligners might indicate a potential alteration of oral health, especially in the initial phase of the therapy. The finding of decreased IL-8 expression after seven days of culture cannot be explained by means of this study and needs to be clarified by further testing.

We further tested potential differences between the different aligner surfaces in their effect on epithelial cells. As the aligners are cut beneath the free gingival margin from the residual thermoformed aligner material during the fully automated production process, the inner/rough surface of the aligners is mainly in contact with papilla tissue and its epithelial layer. In inflammatory conditions such as gingivitis, which can also be seen during aligner treatment [8,55], the inflamed gingiva also comes in contact with aligner material. Therefore, the effects of the rough inner surface are of particular interest. We found some significant differences in various parameters between the inner and outer surfaces of aligners. However, the differences between different aligners’ surfaces were markedly smaller than those induced by aligner material per se. Therefore, the variation in the aligners’ surface structure might have only minor effect on their interaction with the oral epithelium.

Our results are in accordance with Premaraj et al., who demonstrated decreased metabolic activity in oral epithelial cells exposed to Invisalign eluates in saline solution [12]. Regarding cell metabolic activity, Martina et al. reported a slight cytotoxic activity of aligners’ material eluates on human gingival fibroblasts [11], whereas Eliades et al. found no evidence of cytotoxicity on human gingival fibroblasts with Invisalign precursor material [13]. These differences can be explained by the use of different aligner material.

### 4.1. Limitations

Our findings reveal the effects on Invisalign SmartTrack material on oral squamous carcinoma cells in vitro but do not necessarily explain the in vivo situation. Further, cells were cultured in specific media, whereas in the oral cavity the contact between aligners and oral epithelium take place in the presence of saliva. Thus, it is reasonable to establish a well-defined experimental model to depict effects that can be triggered by aligners on oral epithelial cells. It would be interesting in the future to measure and compare the variables, which have been tested in the present study on aligners, with other orthodontic appliances, such as metal or ceramic fixed appliances.

Ca 9-22 cells used in the present study derived from gingival squamous carcinoma and have first been established in 1978 [56]. Although these cells are often used as oral epithelium model for testing various dental materials [57,58], they do not resemble the properties of oral epithelium entirely. Particularly, oral squamous carcinoma cells are less prone to apoptosis than the primary epithelial cells [59]. This aspect is especially crucial for the assessment of the cytotoxic effect of dental materials. Furthermore, tumor transformation of oral epithelium is accompanied by the alteration in the expression of some functional properties, particularly E-cadherin [42]. Therefore, further studies should be performed to evaluate the effects of aligners’ materials on primary oral epithelial cells.

### 4.2. Clinical Relevance

Epithelial parameters are stimulated by aligners’ material which seems to affect oral epithelial cells and could contribute to inflammatory reactions, however, at a very early phase of aligner wear. This finding should be considered if treating patients with impaired oral health conditions such as periodontal disease or gingivitis. It has to be proven in future studies if these effects accumulate during orthodontic treatment, or can be reversed by gradually changing to the next set of aligners. This could be of pivotal interest, as orthodontic treatment in patients with previous periodontal disease should be performed with low forces. This should be acknowledged in treatment planning as an earlier change to the next set of aligners would result in lower forces for each aligner set.

## 5. Conclusions

Our findings show that oral epithelial cells do not grow on Invisalign SmartTrack material. This material inhibits proliferation but has no cytotoxic effects on oral epithelial cells in vitro. Further, aligners’ material influence the expression of various factors involved in the epithelial barrier function and local inflammatory response. The role of these factors in any of the side effects of aligners’ therapy has still to be established in clinical studies.

## Figures and Tables

**Figure 1 materials-13-05311-f001:**
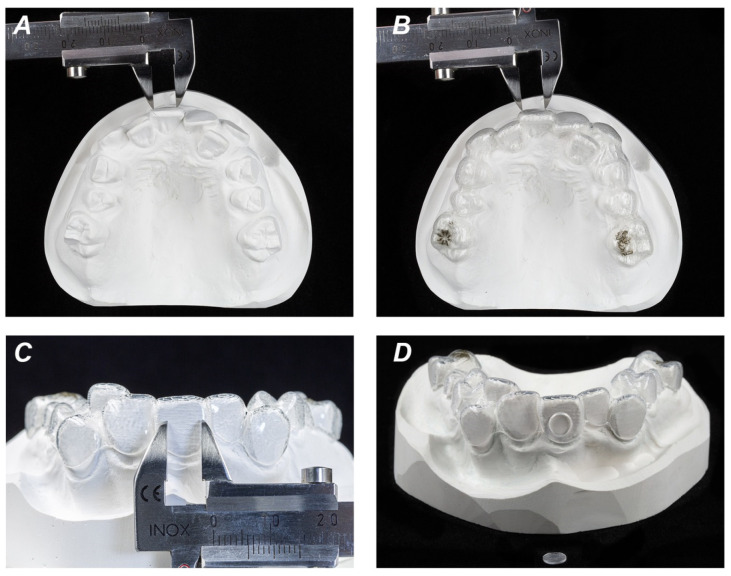
Preparation of orthodontic aligners used in the study. Conventional orthodontic plaster models were ground plane in the region of buccal areas. Required diameter of 6 mm of flat surface was confirmed by a conventional orthodontic caliper (**A**). Flat aligner surface of 6 mm equivalent to buccal plaster cast surface (**B**,**C**). Discs of 6 mm in diameter were cut out of regions with flat crown surfaces and used in cell culture experiments (**D**).

**Figure 2 materials-13-05311-f002:**
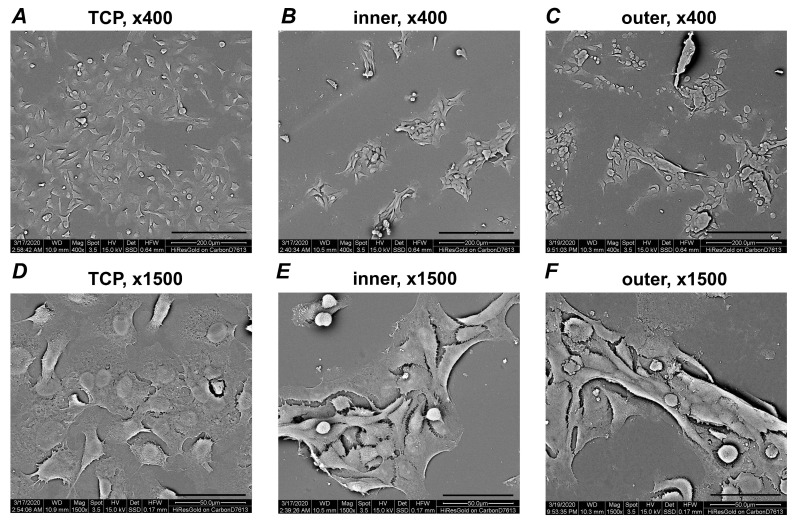
Scanning electron microscopy analysis of Ca9-22 cells grown on different surfaces. Pictures were taken at two different magnifications (upper row, ×400; lower row, ×1500). Cells seeded on TCP (control; **A**,**D**), inner (**B**,**E**) and outer (**C**,**F**) aligners’ surfaces are shown. Pictures were taken after seven days of culture. Scale bars correspond to 50 µm (**A**–**C**) or 200 µm (**D**–**F**).

**Figure 3 materials-13-05311-f003:**
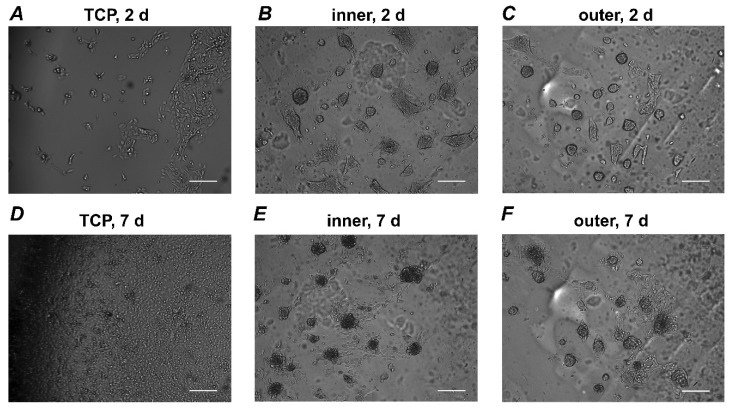
Live cell analyzer pictures. Ca9-22 cells were grown on different surfaces and the pictures were taken every 30 min. Cells seeded on TCP (control; **A**,**D**), inner (**B**,**E**) and outer (**C**,**F**) aligners’ surfaces are shown. Shown pictures were taken after two (**A**–**C**) and seven days (**D**–**F**) of culture. Scale bars correspond to 200 µm.

**Figure 4 materials-13-05311-f004:**
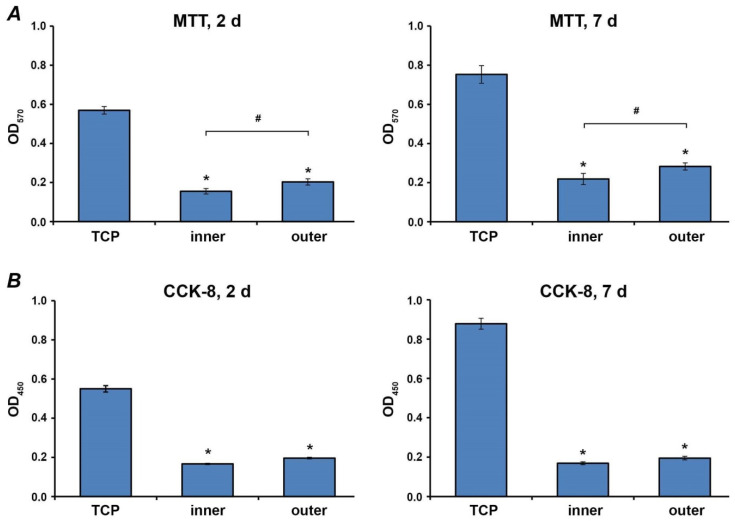
Proliferation/viability of Ca9-22 cells grown on different surfaces. Ca9-22 cells were cultured on TCP, inner and outer aligners’ surfaces. Proliferation/viability was measured after two and seven days incubation by MTT (**A**) and CCK-8 (**B**) assays. Cells grown on tissue culture plastic served as control. Y-axis represents the optical density (OD) measured at 570 nm and 450 nm. Data represents mean ± s.e.m. of four independent experiments. *—significantly lower compared to control; #—significantly different between groups. *p* < 0.05.

**Figure 5 materials-13-05311-f005:**
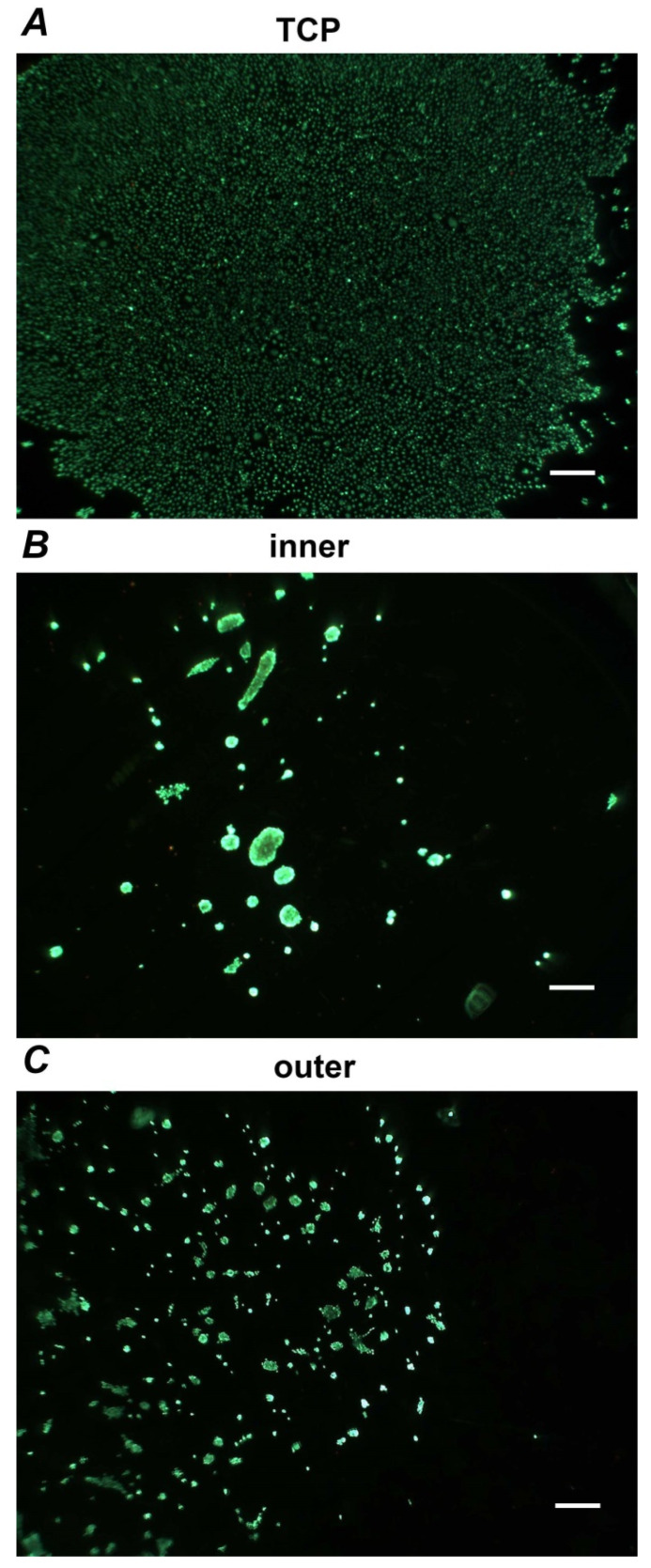
Live/dead Staining. Ca9-22 cells were grown on TCP (**A**), inner (**B**) and outer (**C**) aligners’ surfaces for seven days and stained with Live/dead staining kit. Vital cells are visible as green while the dead cells are presented red. Images are taken from representative experiment. Scale bar correspond to 500 µm.

**Figure 6 materials-13-05311-f006:**
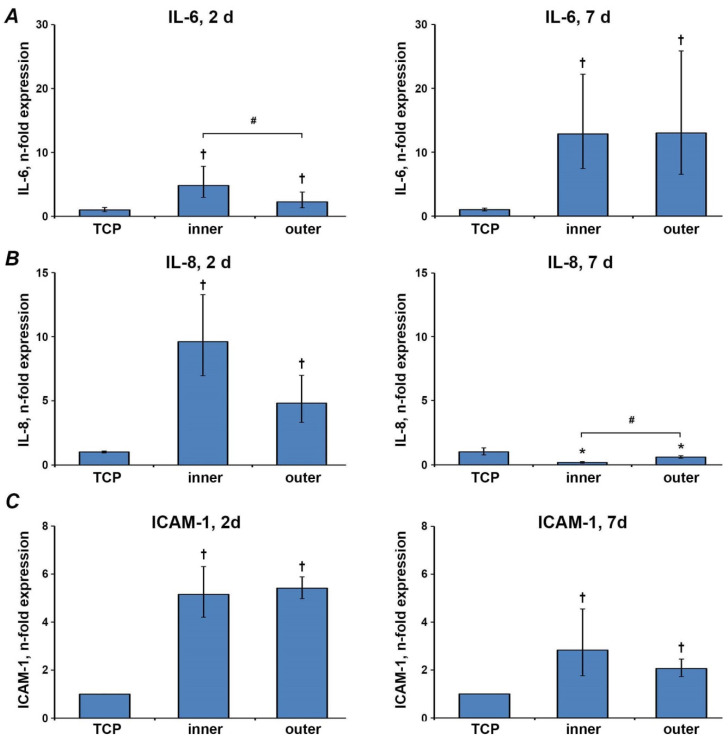
Gene expression of inflammatory markers. Gene expression of IL-6 (**A**), IL-8 (**B**) and ICAM1 (**C**) after two and seven days in Ca9-22 cells grown on different aligners’ surfaces were measured by qPCR. Y-axes represent n-fold expression in relation to Ca9-22 cells grown on TCP (n-fold expression = 1), calculated using 2^−ΔΔCt^ method. Data are presented as the mean ± s.e.m. of four independent experiments. †—significantly higher compared to control; *—significantly lower compared to control; #—significantly different between groups. *p* < 0.05.

**Figure 7 materials-13-05311-f007:**
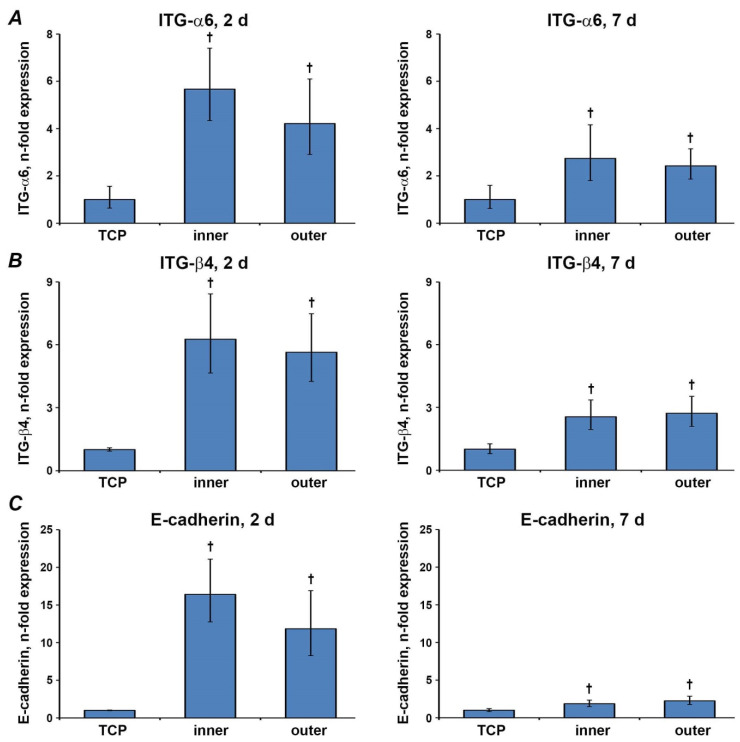
Gene expression of epithelial barrier markers. Gene expression of ITGα-6 (**A**), ITGβ-4 (**B**) and E-cadherin(**C**) after two and seven days in Ca9-22 cells grown on different aligners’ surfaces and TCP were measured by qPCR. Y-axes represent n-fold expression in relation to Ca9-22 cells grown on TCP (n-fold expression = 1), calculated using 2^−ΔΔCt^ method. Data are presented as the mean ± s.e.m. of four independent experiments. † Significantly higher compared to control; *p* < 0.05.

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
