# Peer review of "Behaviour of Human Oral Epithelial Cells Grown on Invisalign® SmartTrack® Material"

_materials, 2020, doi:10.3390/ma13235311_

Round 1

Reviewer 1 Report

I had the opportunity of reviewing this interesting in vitro study about the effects of Invisalign material on oral epithelial cells.

The study is well designed, well performed and well written and merits publication on Materials, I would like to enlighten some points that should be checked or changed.

  1. The “conclusions” in the abstract (even if it is not structured) should be reorganized following the conclusion section of the manuscript. This because I found those conclusions more balanced and appropriate to describe the findings of the study.
  2. The study used an oral squamous carcinoma cell line, are these tumoral cells? Could this alter the results compared with the epithelial cells of healthy oral soft tissue? If so, this should be carefully reported in different parts of the manuscript, especially in the discussion and conclusions.
  3. In the “cell culture” section the authors wrote “Cells seeded by the similar protocol on tissue culture plastic served as control” why do you speak about SIMILAR protocol? Has the protocol some differences for TCP compared with Invisalign material?
  4. Please check all the abbreviations throughout the manuscript and report them the first time extended and with the abbreviation in round brackets.
  5. Try to use a similar figure description for Figure 2 and Figure 3.

Author Response

REVIEWER'S COMMENT

The “conclusions” in the abstract (even if it is not structured) should be reorganized following the conclusion section of the manuscript. This because I found those conclusions more balanced and appropriate to describe the findings of the study.

AUTHORS’ RESPONSE

Thank you for this comment. We reorganized this section in the abstract as followed:

Invisalign Smart Track material exhibits no cytotoxic effect on oral epithelial cells, but alters their behavior and the expression of proteins involved in the inflammatory response, and barrier function. The clinical relevance of these effects has still to be established.

We also shortened the abstract to adjust it to 250 words.

----------------------------------------------------------------------------------

REVIEWER'S COMMENT

The study used an oral squamous carcinoma cell line, are these tumoral cells? Could this alter the results compared with the epithelial cells of healthy oral soft tissue? If so, this should be carefully reported in different parts of the manuscript, especially in the discussion and conclusions.

AUTHORS’ RESPONSE

This is an important aspect. We complemented this aspect by the following in the discussion sections (P. 13, lines 331-333) and also emphasized this important aspect as a potential limitation of our study (P. 14, lines 380-388).

----------------------------------------------------------------------------------

REVIEWER'S COMMENT

In the “cell culture” section the authors wrote “Cells seeded by the similar protocol on tissue culture plastic served as control” why do you speak about SIMILAR protocol? Has the protocol some differences for TCP compared with Invisalign material?

AUTHORS’ RESPONSE

This was an unfortunate wording. Cells were grown on TCP according the same protocol as the experimental group, which means that first, cells were seeded in 15 µl of medium, and the additional 85 µl of medium were added after 4 h. We have improved this section as follows:

As control, 5x103 cells were seeded in 15µl medium on tissue culture plastic (TCP) which was followed by adding further 85µl medium after 4 h.

----------------------------------------------------------------------------------

REVIEWER'S COMMENT

Please check all the abbreviations throughout the manuscript and report them the first time extended and with the abbreviation in round brackets.

AUTHORS’ RESPONSE

We checked the abbreviations and explained them when met first time in the text.

-----------------------------------------------------------------------------------

REVIEWER'S COMMENT

Try to use a similar figure description for Figure 2 and Figure 3.

We adapted Figure 3 according to Figure 2 by the following figure description:

AUTHORS’ RESPONSE

Figure 3. Live cell analyzer pictures. Ca9-22 cells were grown on different surfaces and the pictures were taken every 30 min. Cells seeded on TCP (control; A, D), inner (B, E) and outer (C, F) Invisalign surfaces are shown. Pictures were taken after two days (A, B, C) or seven days (D, E, F) of culture. Scale bars correspond to 200 µm.

Reviewer 2 Report

1. The authors should describe how the cells behave on the curved materials. Considering the buccal and lingual contour of molars,  "flat" and "even" surface of a diameter of 6 mm cannot be achieved. As shown in figure 1. the samples look curved rather than flat. Anatomical evidence of "flatness" should be stated.

2. In the discussion, please explain the cellular response in the aspect of materials' components .

Author Response

REVIEWER COMMENT

  1. The authors should describe how the cells behave on the curved materials. Considering the buccal and lingual contour of molars,  "flat" and "even" surface of a diameter of 6 mm cannot be achieved. As shown in figure 1. the samples look curved rather than flat. Anatomical evidence of "flatness" should be stated.

AUTHORS’ ANSWER

Unfortunately, due to some misunderstandings, the description of the methods used for aligner preparation was misleading and incomplete. We apologize for this mistake. In the present study, we have prepared a special model where some teeth were ground to obtain a flat area. In the revised version, we have improved the description of the aligner preparation process and visualized it in the new Figure 1.

REVIEWER COMMENT

  1. In the discussion, please explain the cellular response in the aspect of materials' components.

AUTHORS’ ANSWER

Thank you for this comment. In the revised version, we have added a paragraph about the potential role of Invisalign material components in cell response (See, P. 12, lines 303-313).

Reviewer 3 Report

Thanks for the opportunity to review this paper, this is interesting research due to seek assess the behavior of human oral epithelial cells grown on
3 Invisalign® material.

This study has its merits because the authors performed several testing, however, the results were already predictable by the hydrophobic behavior of Invisalign SmarTrack material. This reviewer would recommend adding a simple test of contact angle to see the difference of surface energy among the three surfaces. Also, rather than a simple comparison between different materials, researchers are encouraged to look at properties that can be really characterized by a sound experimental model.

Author Response

Reviewer 3

REVIEWER COMMENT

This study has its merits because the authors performed several testing, however, the results were already predictable by the hydrophobic behavior of Invisalign SmarTrack material. This reviewer would recommend adding a simple test of contact angle to see the difference of surface energy among the three surfaces. Also, rather than a simple comparison between different materials, researchers are encouraged to look at properties that can be really characterized by a sound experimental model.

AUTHORS’ ANSWER

Thank you for this comment. In the revised version, we have added the contact angle information (see, P. 5, lines 193-196) and found it to be about the same on all three surfaces. Thus, the differences in cell behavior could be explained by different materials rather than by different hydrophobicity. This assumption is also in agreement with the existing literature data and is discussed in detail in our revised version (see, P. 12, line 308-319).

Round 2

Reviewer 1 Report

All the suggestions were performed so the article can be now accepted for publication.

Author Response

Thank you for your comments and recommendation to accept our manuscript.